# Pleiotropy of polygenic factors associated with focal and generalized epilepsy in the general population

**Costin Leu** [1,2,3]*, **Tom G. Richardson**[4], **Tobias Kaufmann**[5], **Dennis van der Meer**[6], **Ole A. Andreassen**[6], **Lars T. Westlye** [6,7], **Robyn M. Busch**[8,9,10], **George Davey Smith**[4], **Dennis Lal**[1,2,8,11]*

**1** Genomic Medicine Institute, Lerner Research Institute, Cleveland Clinic, Cleveland, OH, United States of America, **2** Stanley Center for Psychiatric Research, Broad Institute of Harvard and M.I.T, Cambridge, MA, United States of America, **3** Department of Clinical and Experimental Epilepsy, Institute of Neurology, University College London, London, United Kingdom, **4** MRC Integrative Epidemiology Unit (IEU), Population Health Sciences, Bristol Medical School, University of Bristol, Oakfield House, Bristol, United Kingdom, **5** Division of Mental Health and Addiction, NORMENT, KG Jebsen Centre for Psychosis Research, Oslo University Hospital & Institute of Clinical Medicine, University of Oslo, Oslo, Norway, **6** Norwegian Centre for Mental Disorders Research, University of Oslo and Oslo University Hospital, Oslo, Norway, **7** Department of Psychology, University of Oslo, Oslo, Norway, **8** Epilepsy Center, Neurological Institute, Cleveland Clinic, Cleveland, OH, United States of America, **9** Department of Psychiatry & Psychology, Neurological Institute, Cleveland Clinic, Cleveland, OH, United States of America, **10** Department of Neurology, Neurological Institute, Cleveland Clinic, Cleveland, OH, United States of America, **11** Cologne Center for Genomics (CCG), University of Cologne, Cologne, Germany

* leuc@ccf.org (CL); lald@ccf.org (DL)

**Data Availability Statement:** The epilepsy GWAS summary statistics underlying the results presented in the study are available from the ILAE Consortium on Complex Epilepsies: http://www.

## Abstract

Epilepsy is clinically heterogeneous, and neurological or psychiatric comorbidities are frequently observed in patients. It has not been tested whether common risk variants for generalized or focal epilepsy are enriched in people with other disorders or traits related to brain or cognitive function. Here, we perform two brain-focused phenome association studies of polygenic risk scores (PRS) for generalized epilepsy (GE-PRS) or focal epilepsy (FE-PRS) with all binary brain or cognitive function-related traits available for 334,310 European-ancestry individuals of the UK Biobank. Higher GE-PRS were associated with not having a college or university degree ($P = 3.00 \times 10^{-4}$), five neuroticism-related personality traits ($P < 2.51 \times 10^{-4}$), and having ever smoked ($P = 1.27 \times 10^{-6}$). Higher FE-PRS were associated with several measures of low educational attainment ($P < 4.87 \times 10^{-5}$), one neuroticism-related personality trait ($P = 2.33 \times 10^{-4}$), having ever smoked ($P = 1.71 \times 10^{-4}$), and having experienced events of anxiety or depression ($P = 2.83 \times 10^{-4}$). GE- and FE-PRS had the same direction of effect for each of the associated traits. Genetic factors associated with GE or FE showed similar patterns of correlation with genetic factors associated with cortical morphology in a subset of the UKB with 16,612 individuals and T1 magnetic resonance imaging data. In summary, our results suggest that genetic factors associated with epilepsies may confer risk for other neurological and psychiatric disorders in a population sample not enriched for epilepsy.

epigad.org/gwas_ilae2018_16loci.html. The study samples are available from the UK Biobank: http://www.ukbiobank.ac.uk/register-apply.

**Funding:** This work was supported by the Integrative Epidemiology Unit, which receives funding from the UK Medical Research Council and the University of Bristol (MC_UU_00011/1). TGR is a UKRI Innovation Research Fellow (MR/S003886/1). RMB received support from the NIH/NCATS, CTSA UL1TR000439, Cleveland, Ohio. The funders had no role in study design, data collection and analysis, decision to publish, or preparation of the manuscript.

**Competing interests:** The authors have declared that no competing interests exist.

## Introduction

More than 50 million people worldwide have epilepsy, making it one of the most common neurological diseases globally (www.who.int). Epilepsy is a heterogeneous neurological disorder characterized by an enduring predisposition to generate epileptic seizures [1]. Apart from seizure type and etiology, an array of neurological and psychiatric comorbidities contribute to the phenotypic heterogeneity of epilepsy [2]. Emerging evidence supports a widespread genetic sharing between all common brain disorders [3], including epilepsy and behavioral-cognitive traits.

Genome-wide association studies (GWAS) for common forms of epilepsy have recently identified common genetic risk variants for epilepsy and the two main epilepsy syndromes; generalized epilepsy (GE) and focal epilepsy (FE). The two main epilepsy syndromes differ by definition in their seizure semiology: FE is characterized by focal seizures originating from one cerebral hemisphere; GE is characterized by seizures involving both cerebral hemispheres [4]. Major depression and anxiety disorders are the most common psychiatric comorbidities, both reported at equivalent frequencies in all individuals with epilepsy (~20%) [5,6], and individuals with drug-resistant epilepsy (~50%) [7]. Epilepsy and comorbid disorders show strong heritability rates in family studies [8]. Subsequently, in recent years, common variants have been identified for epilepsy and virtually all comorbid brain disorders. Considering the substantial polygenicity of complex disorders [9], polygenic risk scores that represent a quantitative measure of an individual's genetic risk towards disease can be used to explore pleiotropic effect between disorders, as shown for schizophrenia [10] and a selection of behavioral traits [11] in otherwise healthy individuals.

In this study, we use polygenic risk scores (PRS) to examine the pleiotropic effects of genetic factors associated with epilepsy using two approaches. First, we study whether PRS for GE or FE are associated with brain or cognitive function-related traits, by conducting a PRS-phenome association study in 334,310 European-ancestry individuals from the UK Biobank (UKB) [12]. Second, we investigate whether GE and FE are correlated at the level of genetic variants associated with brain morphology in a subset of the UKB with 16,612 European-ancestry individuals and T1-weighted magnetic resonance imaging (MRI) data.

## Methods

### UK Biobank resource

The UKB is an open-access population-based prospective study with over 500,000 participants aged 40–69 years who were recruited in 2006–2010 [13]. Phenotypic data for >2000 traits & disorders were collected from questionnaires, physical measures, sample assays, accelerometry, multimodal imaging (e.g., MRI), and health records. All individuals of the UKB are genotyped with the Axiom array [12]. Of note, despite efforts to ensure a broad distribution across all health outcomes, there is evidence of enrichment of the UKB with healthy volunteers [14]. The healthy volunteer selection bias is likely to be particularly strong for mental disorders, where disorder status or symptoms may influence participation in research [15]. For this study, we used all individuals of European ancestry of the UKB that had genotypic data (N = 334,310).

### PRS-phenome association

We made use of the rich variety of phenotypic information in the UKB [12] to determine if GE- and FE-PRS are associated with traits and disorders related to any brain or cognitive function, using a phenome-association method described by Richardson et al. (2019) [16]. Out of all UKB phenotypes (>2000), we found 42 brain or cognitive function-related phenotypes (i.e.

neurological, neurodegenerative, and psychiatric disorders, personality traits, and educational attainment) that were binary and heritable ($P<0.05$ in LD score regression as calculated previously: https://nealelab.github.io/UKBB_ldsc/). Focusing on binary phenotypes (i.e. excluding continuous and ordinal phenotypes) enabled comparison of the final effect estimates across all tested traits.

We then generated polygenic risk scores for generalized (GE-PRS) and focal epilepsy (FE-PRS) for all UKB individuals using single-nucleotide polymorphism (SNP) weights derived from summary statistics of the International League Against Epilepsy (ILAE) Consortium on Complex Epilepsies GWAS for GE and FE [17]. The SNPs were pruned based on $P<0.5$. All remaining SNPs were extracted from the UKB imputed genotype data based on the Haplotype Reference Consortium r1.1 [18] and 1000 genomes [19] phase 3 reference panels. When a GWAS SNP was not present in the UKB genotype data, we attempted to identify a proxy SNP in high linkage disequilibrium with r2≥0.8. The remaining SNPs were clumped to a subset of weakly correlated SNPs (based on r2<0.1 within 500kb from the SNP with the lowest $P$-value at each locus). PRS for 334,310 individuals enrolled in the UKB study were generated using the allelic scoring function, as implemented in PLINKv1.9 [20]. Individual PRS were calculated as the sum of weighted effect alleles divided by the number of SNPs in the analysis. We excluded all individuals that were included in the GWAS studies used for PRS development using KING (kinship coefficient >0.0442) [21].

We used logistic regression, adjusted for sex, and the first four principal components of ancestry (PCs) to test for association between GE- or FE-PRS and the 42 brain or cognitive function-related traits of the UKB. We selected only four PCs, following the example in Khera et al. (2018) [22], and to avoid overfitting when too many PCs are included association model [23,24]. The threshold to reject the null hypothesis was set to $\alpha = 5.95 \times 10^{-4}$, after the Bonferroni correction method for multiple testing (2x42 tests).

## Vertex-wise genetic correlation

Vertex-wise genetic correlation maps with brain morphology were built following the methods presented by Kaufmann et al. (2018) [25]. Cortical reconstruction was performed in a UKB [12] subset of 16,612 predominantly healthy [26] individuals of European ancestry with T1-weighted MRI data, using Freesurfer [27]. Of note, the impact of confounding factors such as comorbid disorders or medication on cortical morphology was reduced by the MRI data acquisition from predominantly healthy participants in the UKB [25]. We computed surface maps for cortical thickness and cortical surface area, registered to *fsaverage4* space (2,562 vertices), smoothed using a kernel with full width at half maximum of 15 mm. GWASs of each vertex's thickness and area were performed using PLINKv1.9 [20], with adjustment for age, $age^2$, sex, scanning site, and the first four genetic principal components of ancestry. Next, we estimated the genetic correlation between each vertex's thickness and surface area and the GWAS summary statistics for GE and FE [17], using LD-score regression [28]. The resulting vertex-wise genetic correlation maps of cortical morphology with epilepsy were tested for correlation with each other using Spearman correlation. Statistical testing of the correlations was performed using spin-rotation based permutation testing with 10,000 permutations [29].

## Results

### Epilepsy-PRS are associated with brain or cognitive function-related traits

We found seven traits associated with GE-PRS (surpassing Bonferroni correction for 2x42 tests, a = $5.95 \times 10^{-4}$): not having a college or university degree ($P = 3.00 \times 10^{-4}$; Fig 1), five personality traits associated with neuroticism (sensitivity / hurt feelings, $P = 1.65 \times 10^{-6}$; worry too

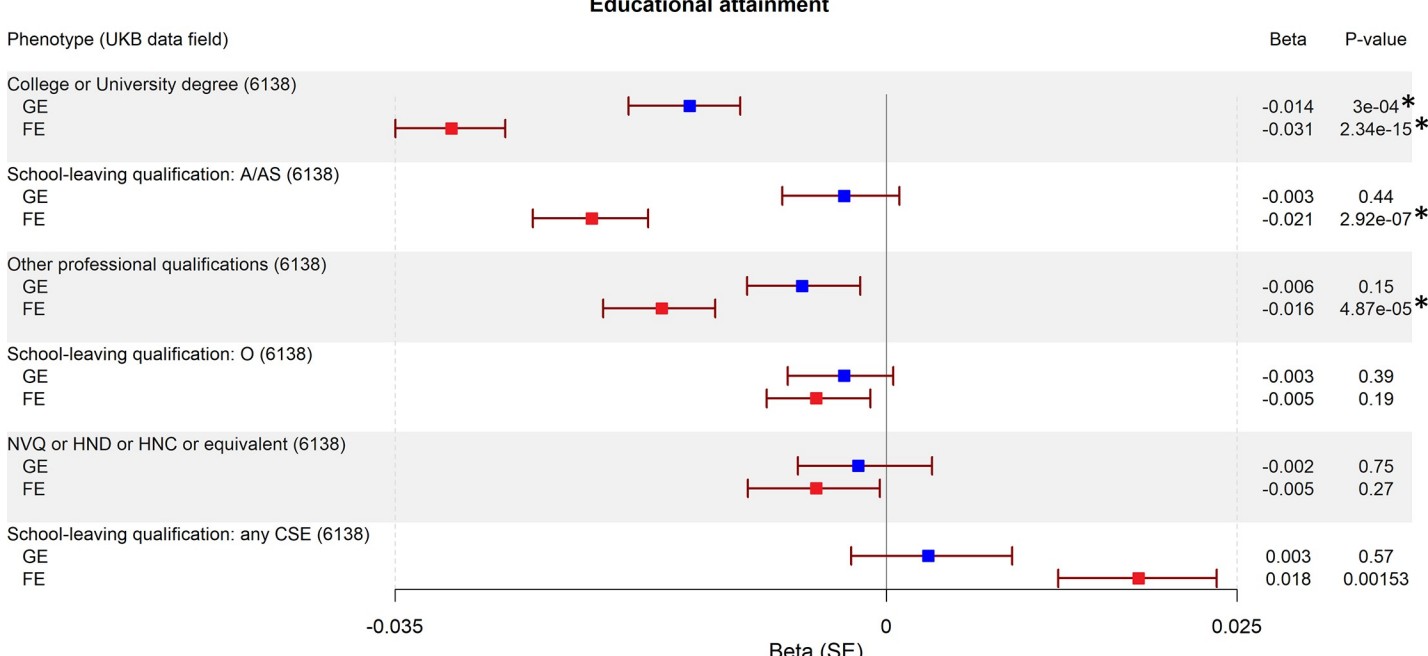

**Fig 1. Association between genetic risk for GE and FE and educational attainment phenotypes.** Plotted are the PRS-phenome association results for GE and FE for all binary educational attainment traits in the UKB. The betas for GE-PRS are highlighted in blue, and for FE-PRS in red. *P*-values were calculated using a logistic regression model, adjusted for sex, and the first four principal components of ancestry. The threshold to reject the null hypothesis was set to $\alpha = 5.95 \times 10^{-4}$ after Bonferroni correction for 84 tests. Legend: UKB: UK Biobank, A/AS: British General Certificate of Education (GCE) advanced/advanced supplementary level equivalent to the US associate degree, O: GCE lowest pass grade at ordinary level, NVQ: National Vocational Qualifications / British work-based awards, HND: British Higher National Diploma, HNC: British Higher National Certificate, SE: standard error, *: *P*-value is surpassing the Bonferroni corrected threshold to reject the null hypothesis.

long after embarrassment, $P = 1.24 \times 10^{-5}$; mood swings, $P = 2.27 \times 10^{-5}$; miserableness, $P = 2.27 \times 10^{-5}$; worrier / anxious feelings, $P = 2.51 \times 10^{-4}$; Fig 2), and having ever smoked ($P = 1.27 \times 10^{-6}$; Fig 3). The largest effect sizes for high GE-PRS were observed in association with neuroticism-related personality traits and smoking. In a separate analysis, we found six traits associated with high FE-PRS: several measures indicating low educational attainment (not having a college or university degree, $P = 2.34 \times 10^{-15}$; not having a General Certificate of Education Level A/AS [UK equivalent of a US associate degree], $P = 2.92 \times 10^{-7}$; not having any other professional qualifications, $P = 4.87 \times 10^{-5}$; Fig 1), one personality trait associated with neuroticism (mood swings, $P = 2.33 \times 10^{-4}$; Fig 2), having ever smoked ($P = 1.71 \times 10^{-4}$; Fig 3), and having experienced events of anxiety or depression ($P = 2.83 \times 10^{-4}$; Fig 4). The largest effect sizes for high FE-PRS were observed in association with low educational attainment. Out of the 42 traits related to cognitive functioning, seven showed association with either GE-PRS or FE-PRS and three with both types of epilepsy-PRS. Notably, the PRS had the same direction of effect for all ten traits associated with GE- or FE-PRS, including the not associated epilepsy subtype. Additional not associated traits are shown in the supplementary material (S1 Fig–S3 Fig).

**GE and FE show similar patterns of genetic correlation at vertex-level with cortical thickness and cortical surface area.** We used a recent cortico-genetic mapping approach [25] to explore whether GE and FE are genetically correlated at the level of brain morphology. We performed vertex-level genetic correlation analyses between epilepsy (GE and FE) and cortical surface area and thickness. The genetic correlation results for each vertex were mapped onto the brain. Each map reflects the overlap between the genetic architectures of cortical morphology at the vertex level with each of the two epilepsy syndromes (Fig 5). The maps for GE

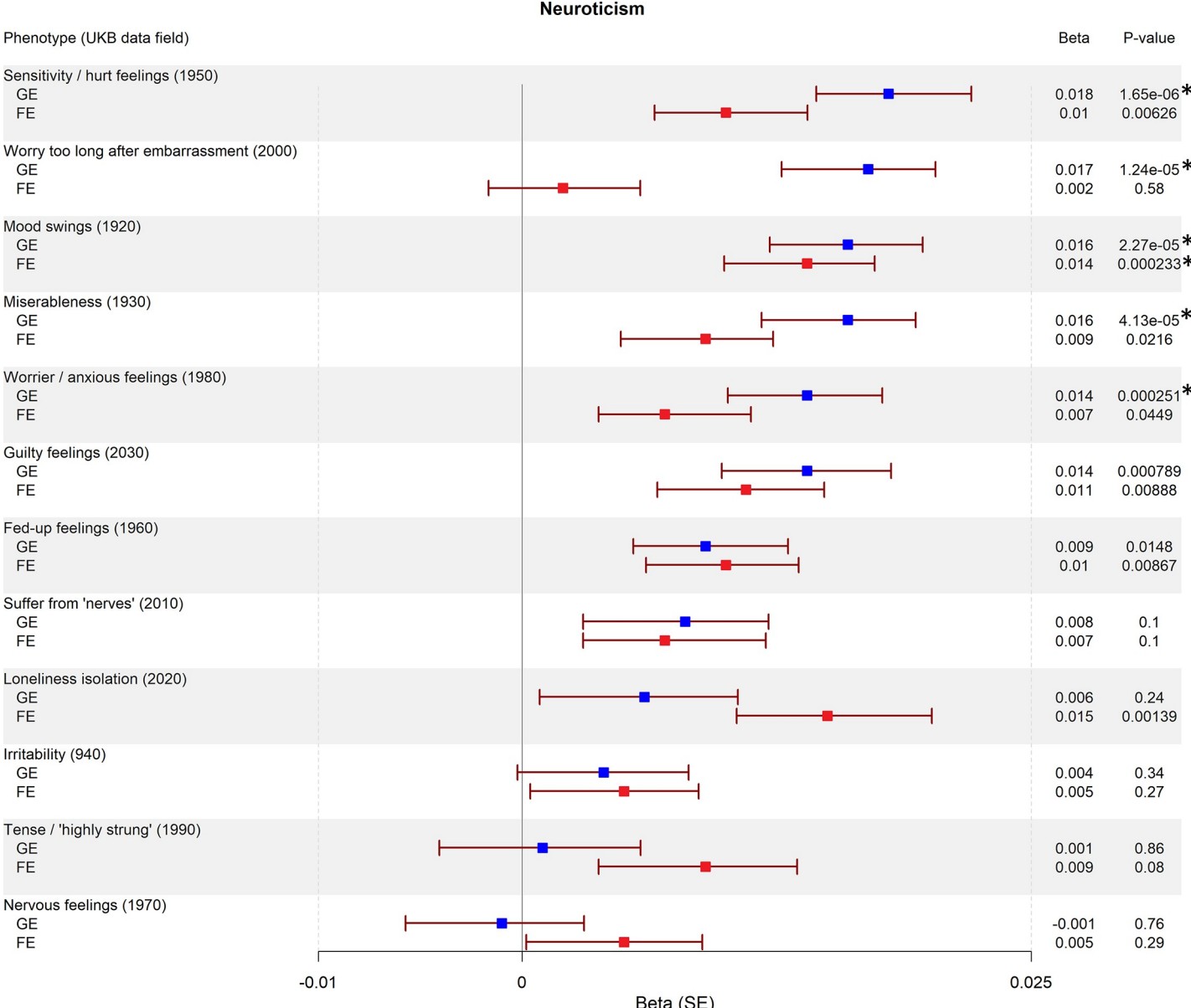

**Fig 2. Association between genetic risk for GE and FE and neuroticism-related traits.** Plotted are the PRS-phenome association results for GE and FE for all binary neuroticism-related traits in the UKB. The betas for GE-PRS are highlighted in blue, and for FE-PRS in red. *P*-values were calculated using a logistic regression model, adjusted for sex, and the first four principal components of ancestry. The threshold to reject the null hypothesis was set to α = 5.95x10$^{-4}$ after Bonferroni correction for 84 tests. Legend: UKB: UK Biobank, SE: standard error, *: *P*-value is surpassing the Bonferroni corrected threshold to reject the null hypothesis.

were weakly correlated with the maps for FE for both parameters: cortical thickness (correlation coefficient rho = 0.24, $P<10^{-4}$, Fig 5C) and cortical surface area (correlation coefficient rho = 0.17, $P = 4x10^{-4}$, Fig 5D).

## Discussion

We investigated the role of genetic risk for GE and FE across 42 brain or cognitive function-related traits in 334,310 individuals of the UKB cohort. Our results suggest that high epilepsy PRS are associated with low educational attainment, personality traits associated with

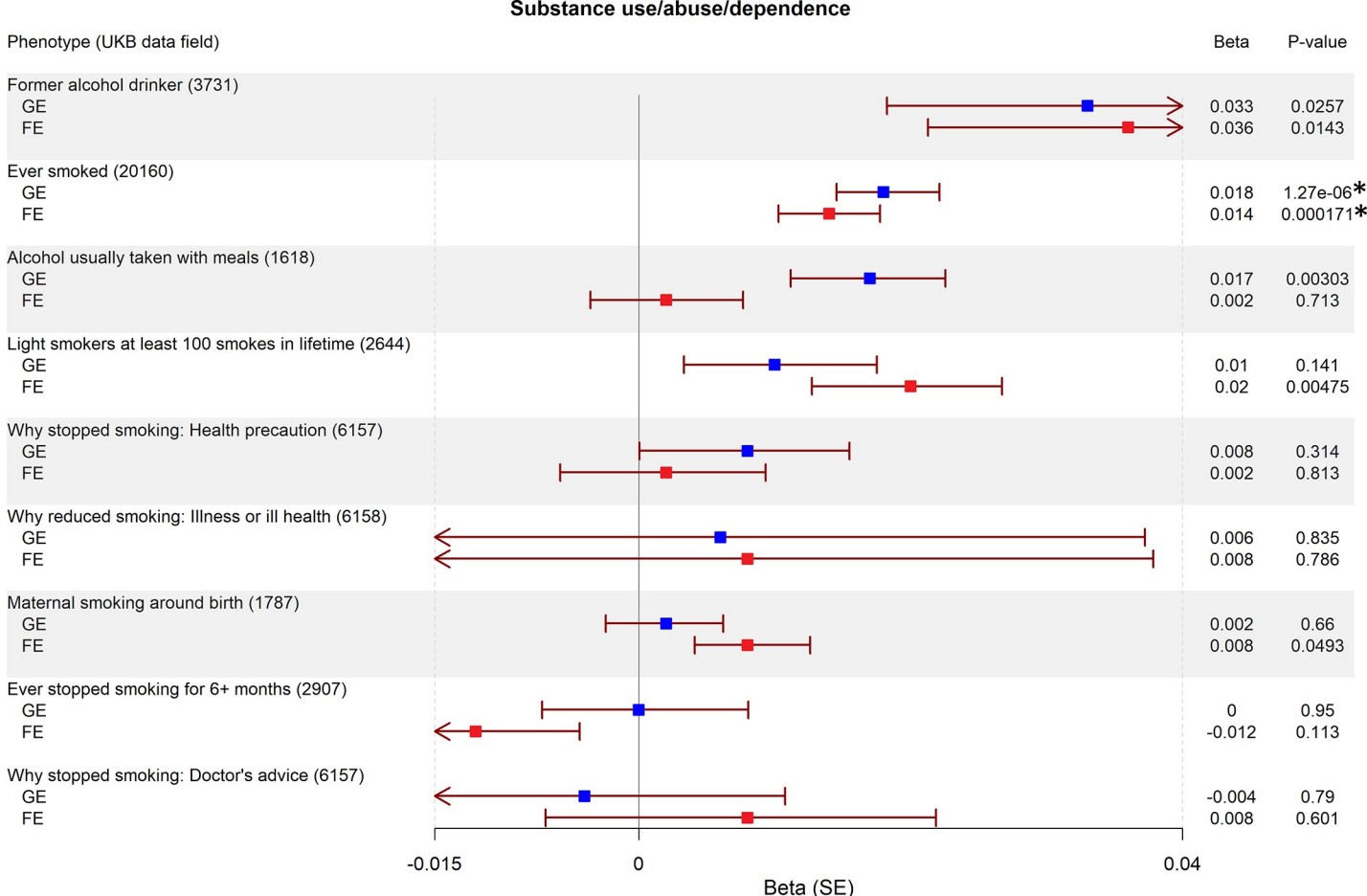

**Fig 3. Association between genetic risk for GE and FE and mental and behavioral disorders due to substance abuse.** Plotted are the PRS-phenome association results for GE and FE for all binary mental and behavioral disorders due to psychoactive substance abuse in the UKB. The betas for GE-PRS are highlighted in blue, and for FE-PRS in red. *P*-values were calculated using a logistic regression model, adjusted for sex, and the first four principal components of ancestry. The threshold to reject the null hypothesis was set to $\alpha = 5.95 \times 10^{-4}$ after Bonferroni correction for 84 tests. Legend: UKB: UK Biobank, SE: standard error, *: *P*-value is surpassing the Bonferroni corrected threshold to reject the null hypothesis.

neuroticism, and smoking behavior in a population sample not enriched for epilepsy. Our results confirm the genetic correlation between intelligence (as a proxy for educational attainment) and epilepsy, which is the only genetic correlation of epilepsy shown as significant in several studies [3,17]. The genetic pleiotropy between epilepsy and neuroticism observed in this study was missed in a larger study that employed linkage disequilibrium (LD) score regression [3], showing the value of examining pleiotropy with different methods. Currently, there is no single method that performs best for all possible trait pairs and study parameters [30]. Our novel findings are in line with previous evidence of neuroticism being genetically correlated to educational attainment [31] and major depression disorder [32], one of the most frequent comorbidities in individuals with epilepsy [5]. In addition, neuroticism is phenotypically correlated with smoking behavior [33]. Out of ten traits associated with either GE-PRS or FE-PRS, three were associated with both types of epilepsy-PRS. GE- and FE-PRS had the same direction of effect for all ten traits, including the not associated epilepsy subtype. To further explore the relationship between GE and FE, we investigated the correlation between maps of cortical morphologies of GE and FE, derived from correlation analyses with genetic factors

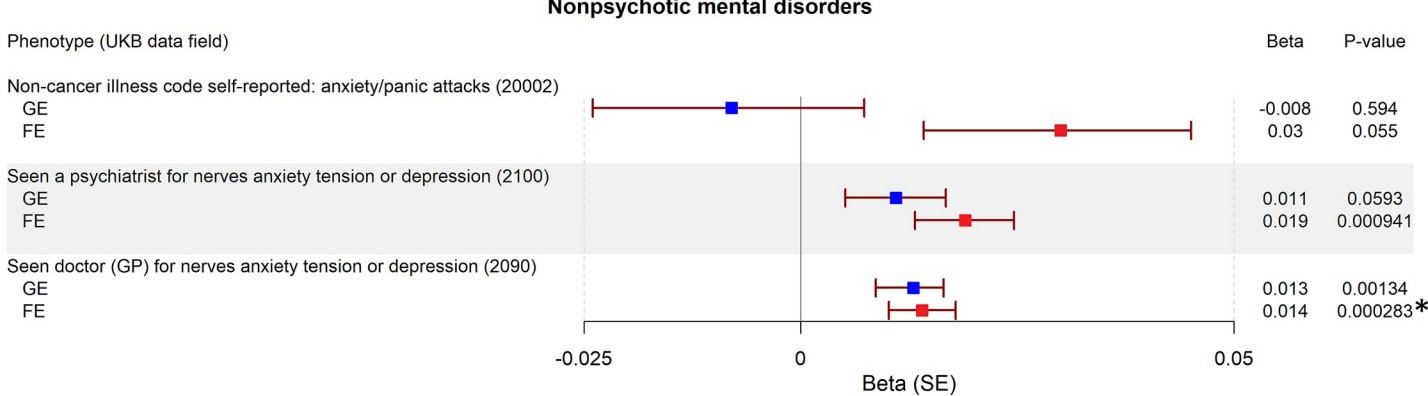

**Fig 4. Association between genetic risk for GE and FE and nonpsychotic mental disorders.** Plotted are the PRS-phenome association results for GE and FE for all binary nonpsychotic mental disorders in the UKB. The betas for GE-PRS are highlighted in blue, and for FE-PRS in red. *P*-values were calculated using a logistic regression model, adjusted for sex, and the first four principal components of ancestry. The threshold to reject the null hypothesis was set to $\alpha = 5.95 \times 10^{-4}$ after Bonferroni correction for 84 tests. Legend: UKB: UK Biobank, SE: standard error, *: *P*-value is surpassing the Bonferroni corrected threshold to reject the null hypothesis.

associated with cortical morphology in a subset of 16,612 UKB individuals with T1 MRI data. The cortico-genetic mapping showed weak correlations between the genetic architecture of cortical morphology of GE with the genetic architecture of cortical morphology of FE. These results are in line with a recent large epilepsy structural brain imaging study, which identified distinct and also shared brain abnormalities in individuals with different epilepsy syndromes [34].

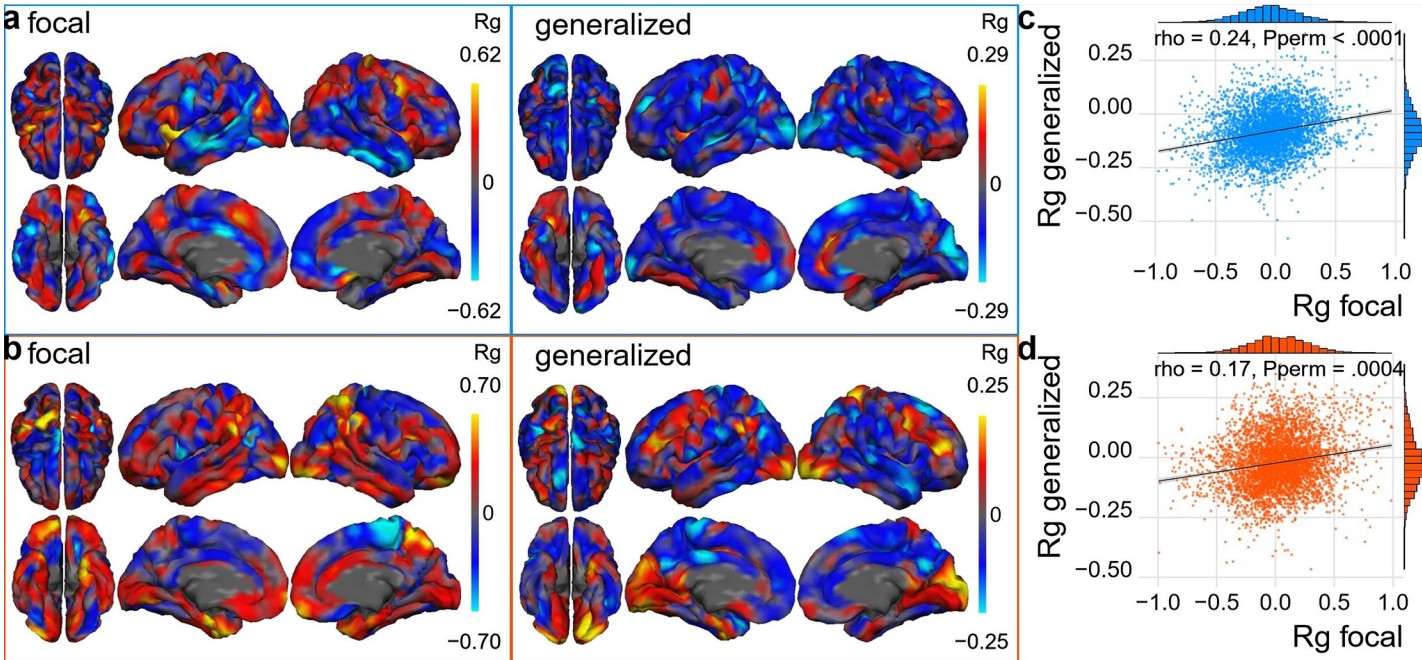

**Fig 5. Vertex-wise genetic correlation between cortical morphology and epilepsy.** (a) Genetic correlation between GWAS summary statistics for cortical thickness and focal or generalized epilepsy at the vertex level. (b) Same as panel a, but for cortical surface area. (c) Spearman correlation between the two cortical thickness maps from panel a. (d) Spearman correlation between the two cortical area maps from panel b. Legend: Rg: genetic correlation, rho: Spearman's correlation coefficient, Pperm: *P*-value after 10k permutations.

The use and generation of PRS is a rapidly developing field, with no established best-practice method. We selected the LD-clumping and *P*-value thresholding method to generate PRS based on a known excellent performance in neurological and psychiatric disorders [35–37]. A commonly used alternative is LDpred, a method that accounts for LD between SNPs and thus allows joint modeling with the potential of improvements in the prediction power [38]. However, the most significant improvements by joint modeling tend to be in diseases with multiple variants in LD that have independent effects (e.g., multiple sclerosis, rheumatoid arthritis, type I diabetes with associated HLA variants) [39]. In epilepsy, variants with independent effects within one LD region have not been demonstrated [17]. Our results should be interpreted in light of PRS generated from GWASs with a strong European bias. Therefore, our analyses are restricted to individuals of European descent, and the generalizability to individuals of non-European ancestry remains to be determined. Although our analyses comprise a large testing cohort for epilepsy PRS, additional phenotypes may uncover other association leads in future larger-scale phenome association studies.

Our PRS-phenome approach illustrates new possibilities for the shared genetic basis of epilepsy and comorbid traits. Comorbidities in epilepsy are common but poorly understood [40]. The quality of life of individuals with epilepsy is reduced by comorbid conditions that include neurological and psychiatric disorders. Possibly, genetic factors associated with epilepsy may contribute directly to the neurological and psychiatric comorbidities observed in individuals with or without epilepsy. Future research is needed to deepen our understanding of pleiotropic effects shared between epilepsy and the various neurological and psychiatric traits.

Using genetics to dissect the heterogeneous clinical representation of individuals with epilepsy represents a new research area. Potentially, polygenic risk for epilepsy includes genetic factors that predispose to a general vulnerability for altered brain function, which is shared with epilepsy comorbid disorders. Our results provide an initial indication of the opportunities and limitations using PRS research in epilepsy. The ongoing growth of large-scale hospital and nation-wide biobanks, which generate genetic data and collect clinical data, will set the stage for future well-powered studies dissecting the interplay of genetic and environmental factors in the etiology of epilepsy and related disorders.

## Supporting information

**S1 Fig. Association between genetic risk for GE and FE and mood disorders.** Plotted are the PRS-phenome association results for GE and FE for all binary mood affective disorders in the UKB. The betas for GE-PRS are highlighted in blue, and for FE-PRS in red. *P*-values were calculated using a logistic regression model, adjusted for sex and the first four principal components of ancestry. The threshold to reject the null hypothesis was set to $\alpha = 5.95 \times 10^{-4}$ after Bonferroni correction for 84 tests. Legend: UKB: UK Biobank, SE: standard error. (DOCX)

**S2 Fig. Association between genetic risk for GE and FE and other diseases of the nervous system.** Plotted are the PRS-phenome association results for GE and FE for all binary diseases of the nervous system in the UKB. The betas for GE-PRS are highlighted in blue, and for FE-PRS in red. *P*-values were calculated using a logistic regression model, adjusted for sex and the first four principal components of ancestry. The threshold to reject the null hypothesis was set to $\alpha = 5.95 \times 10^{-4}$ after Bonferroni correction for 84 tests. Legend: UKB: UK Biobank, SE: standard error. (DOCX)

**S3 Fig. Association between genetic risk for GE and FE and adult personality / behavior disorders.** Plotted are the PRS-phenome association results for GE and FE for all binary adult personality / behavior disorders in the UKB. The betas for GE-PRS are highlighted in blue, and for FE-PRS in red. *P*-values were calculated using a logistic regression model, adjusted for sex and the first four principal components of ancestry. The threshold to reject the null hypothesis was set to $\alpha = 5.95\text{x}10^{-4}$ after Bonferroni correction for 84 tests. Legend: UKB: UK Biobank, SE: standard error.
(DOCX)

## Acknowledgments

This research has been conducted using the UK Biobank Resource under Application Numbers 8786, 15825, and 27412.

## Author Contributions

**Conceptualization:** Costin Leu, Dennis Lal.

**Data curation:** Costin Leu, Tom G. Richardson, Tobias Kaufmann, Dennis van der Meer.

**Formal analysis:** Tom G. Richardson, Tobias Kaufmann, Dennis van der Meer.

**Methodology:** Tom G. Richardson, Tobias Kaufmann, Dennis van der Meer.

**Project administration:** Costin Leu.

**Resources:** Ole A. Andreassen, Lars T. Westlye, George Davey Smith, Dennis Lal.

**Supervision:** Costin Leu, Dennis Lal.

**Visualization:** Costin Leu, Tobias Kaufmann.

**Writing – original draft:** Costin Leu, Dennis Lal.

**Writing – review & editing:** Costin Leu, Tom G. Richardson, Tobias Kaufmann, Dennis van der Meer, Ole A. Andreassen, Lars T. Westlye, Robyn M. Busch, George Davey Smith, Dennis Lal.

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
