## [Decision Letter · Decision Letter 0]

3 Feb 2020

PONE-D-19-34427

Pleiotropy of polygenic factors associated with focal and generalized epilepsy in the general population

PLOS ONE

Dear Dr. Leu,

Thank you for submitting your manuscript to PLOS ONE. After careful consideration, we feel that it has merit but does not fully meet PLOS ONE’s publication criteria as it currently stands. Therefore, we invite you to submit a revised version of the manuscript that addresses the points raised during the review process.

We would appreciate receiving your revised manuscript by Mar 19 2020 11:59PM. To enhance the reproducibility of your results, we recommend that if applicable you deposit your laboratory protocols in protocols.io, where a protocol can be assigned its own identifier (DOI) such that it can be cited independently in the future. For instructions see: http://journals.plos.org/plosone/s/submission-guidelines#loc-laboratory-protocols

We look forward to receiving your revised manuscript.

Kind regards,

Giuseppe Biagini, MD

Academic Editor

PLOS ONE

Journal Requirements:

Reviewers' comments:

Reviewer's Responses to Questions

**Comments to the Author**

1. Is the manuscript technically sound, and do the data support the conclusions?

Reviewer #1: Partly

Reviewer #2: Partly

2. Has the statistical analysis been performed appropriately and rigorously? 

Reviewer #1: I Don't Know

Reviewer #2: Yes

3. Have the authors made all data underlying the findings in their manuscript fully available?

Reviewer #1: Yes

Reviewer #2: Yes

4. Is the manuscript presented in an intelligible fashion and written in standard English?

Reviewer #1: Yes

Reviewer #2: Yes

5. Review Comments to the Author

Reviewer #1: This is the report of a wide retrospective study of pleiotropy of poligenic factors associated with focal and generalized epilepsy. The authors used data from an existing biobank including 334,310 individuals of European ancestry and found high polygenic risk scores for both epilepsies, with generalized epilepsies showing the largest effect sizes for personality traits associated with neuroticism and smoking and focal epilepsies inversely associated with education. Genetic factors associated with the two epilepsies showed similar patterns of correlation with the genetics of cortical morphology in a subsample (16,612 individuals).

The study findings are interesting and add to the existing knowledge on the complexity of the association between genetic and environmental factors in epilepsy. I have, however, a few major queries for the authors to address to provide convincing evidence on these purported associations. More specifically:

1. There is paucity of details on the individuals included in the biobank; the purposes of the biobank are not illustrated and there are no data on the way the information on each individual was collected; the Methods should be expanded to include these details;

2. The diagnosis of focal vs generalized epilepsy has been made according to the 2017 ILAE Classification; however, there is no indication of the criteria used and applied; the validity of this information must be also illustrated;

3. As specified by the authors, “… individuals with a diagnosed brain disorder were excluded.” These exclusion criteria should be specified to put focal epilepsy (largely attributable to structural CNS injuries) in a more correct perspective.

Minor points:

1. Introduction, lines 75 and 76: The sentence in question should be accompanied by a reference;

2. Methods, line 113: SNP and ILAE should be spelled out;

3. Methods, line 116: “a SNP” should perhaps read “an SNP”;

4. Results, lines 145-147: The two sentences are a repetition with what said in the Methods and should be deleted;

5. Results, line 215: “parameter” should read “parameters”;

6. References, lines 331 and 360: Is the journal’s name correct?

Reviewer #2: The authors present interesting analyses that demonstrate pleiotropic effects at the genome-wide level between PRS for generalized and focal epilepsy and a number of behavioral/cognitive traits. These analyses are valuable because they further our understanding of how genetic risk for epilepsy may play a role in other brain or cognitive phenotypes. In general, the article is well written, and analyses are performed to a high technical standard. However, I have a few major and minor comments that I would like addressed before I can recommend the article for publication.

Comments (in no particular order):

1. Can the authors distinguish their work more from the Brainstorm Consortium paper in Science? This paper also looks at genetic correlations between focal and generalized epilepsy and a number of phenotypes. Some of the results overlap with this paper (e.g., educational attainment), and some do not (e.g., neuroticism). The authors are using different methods to examine pleiotropy (PRS vs. LDSC regression), but it seems like results between the papers should overlap more. Along these lines, what are the advantages of using PRS over LDSC regression to examine pleiotropic effects?

2. Can the authors provide evidence that their results are robust to other clumping/pruning parameters or to the use of PRS created using LD Pred? There is no standard in the field for best methods to use when creating PRSs so I think some robustness checks along these lines are needed to show that the results are not sensitive to how the PRS were constructed.

3. Why do the authors only include 4 PCs in their analysis? Are the results robust to the inclusion of more PCs (10 seems to be more common in the literature)?

4. Why do the authors control for sex but not age?

5. In the figures that display the results, it would be helpful for the reader if the results that pass Bonferroni correction are noted in the figure somehow.

6. Can the authors test whether their findings are due to biological or mediated pleiotropy by controlling for whether or not an individual has seizures or has been diagnosed with epilepsy? For example, if correlations between the PRS and educational attainment are not significant after controlling for the epilepsy phenotype, this suggests mediated pleiotropy (e.g., see Schmitz et al., 2019). I think this is useful for clarifying whether or not the genetic risk is independent or working through the phenotype. Particularly in the case of educational attainment, an individual may not have completed much schooling because they struggle with seizures, rather than because genetic variants are independently affecting both phenotypes. Same for neuroticism--if an individual has epilepsy, they may be worried about have seizures in dangerous or inappropriate places and isolate themselves more as a result.

7. The PRS analyses are interesting, but they leave me wondering what genes may be driving these associations. Can the authors test for pleiotropic effects at the gene level for a few key epilepsy genes that have been implicated in the literature? In particular, the PRS results do not tell us much about the underlying biology behind the associations that may be useful from a clinical standpoint. In addition, even within ancestry, the weights from GWAS used to construct the PRS are likely biased by population stratification, assortative mating, or differences in environmental variance between groups (e.g., see Mostafavi et al., 2020). Because of these limitations, analyses at the gene region level would substantially enhance the findings reported in this paper.

Cites:

Mostafavi et al., 2020. Variable prediction accuracy of polygenic scores within an ancestry group eLife; 9:e48376.

Schmitz et al., 2019. Examining sex differences in pleiotropic effects for depression and smoking using polygenic and gene-region aggregation techniques. American Journal of Medical Genetics Part B: Neuropsychiatric Genetics, 180B: 448-468.

6. PLOS authors have the option to publish the peer review history of their article (what does this mean?). If published, this will include your full peer review and any attached files.

Reviewer #1: No

Reviewer #2: No

---

## [Author Response · Author response to Decision Letter 0]

19 Mar 2020

Response to the reviewers provided in the uploaded file "EPI-PRS-UKB_Response_to_Reviewers_March19.docx".

---

## [Decision Letter · Decision Letter 1]

13 Apr 2020

Pleiotropy of polygenic factors associated with focal and generalized epilepsy in the general population

PONE-D-19-34427R1

Dear Dr. Leu,

We are pleased to inform you that your manuscript has been judged scientifically suitable for publication and will be formally accepted for publication once it complies with all outstanding technical requirements.

With kind regards,

Giuseppe Biagini, MD

Academic Editor

PLOS ONE

Additional Editor Comments (optional):

Reviewers' comments:

Reviewer's Responses to Questions

**Comments to the Author**

1. If the authors have adequately addressed your comments raised in a previous round of review and you feel that this manuscript is now acceptable for publication, you may indicate that here to bypass the “Comments to the Author” section, enter your conflict of interest statement in the “Confidential to Editor” section, and submit your "Accept" recommendation.

Reviewer #1: All comments have been addressed

Reviewer #2: All comments have been addressed

2. Is the manuscript technically sound, and do the data support the conclusions?

Reviewer #1: Yes

Reviewer #2: Yes

3. Has the statistical analysis been performed appropriately and rigorously? 

Reviewer #1: Yes

Reviewer #2: Yes

4. Have the authors made all data underlying the findings in their manuscript fully available?

Reviewer #1: Yes

Reviewer #2: Yes

5. Is the manuscript presented in an intelligible fashion and written in standard English?

Reviewer #1: Yes

Reviewer #2: Yes

6. Review Comments to the Author

Reviewer #1: None. All queries have been addressed. There are no remaining concerns to address on my side. The manuscript as revised is to me acceptable for publication.

Reviewer #2: (No Response)

7. PLOS authors have the option to publish the peer review history of their article (what does this mean?). If published, this will include your full peer review and any attached files.

Reviewer #1: No

Reviewer #2: No

---

## [Editor Report · Acceptance letter]

17 Apr 2020

PONE-D-19-34427R1 

Pleiotropy of polygenic factors associated with focal and generalized epilepsy in the general population 

Dear Dr. Leu:

I am pleased to inform you that your manuscript has been deemed suitable for publication in PLOS ONE. Congratulations! Your manuscript is now with our production department. 

With kind regards,

on behalf of

Dr. Giuseppe Biagini 

Academic Editor

PLOS ONE